



**Do alternative inventories converge on the spatiotemporal representation of spring**
**ammonia emissions in France?**
Audrey Fortems-Cheiney[1,*], Gaëlle Dufour[1], Karine Dufossé[2,**], Florian Couvidat[3], Jean-
Marc Gilliot[2], Guillaume Siour[1], Matthias Beekmann[1], Gilles Foret[1], Frederik Meleux[3],
Lieven Clarisse[4], Pierre-François Coheur[4], Martin Van Damme[4], Cathy Clerbaux[4,5] and
Sophie Génermont[2]
[1]Laboratoire Interuniversitaire des Systèmes Atmosphériques, UMR CNRS 7583, Université
Paris Est Créteil et Université de Paris, Institut Pierre Simon Laplace, Créteil, France.
[2]Université Paris-Saclay, INRAE, AgroParisTech, UMR ECOSYS,78850Thiverval-Grignon,
France.
[3]Institut National de l'Environnement Industriel et des Risques, INERIS, 60550 Verneuil en
Halatte, France.
[4]Université libre de Bruxelles, Spectroscopy, Quantum Chemistry and Atmospheric Remote
Sensing (SQUARES), Brussels, Belgium.
[5]LATMOS/IPSL, Sorbonne Université, UVSQ, CNRS, Paris, France.
*now at Laboratoire des Sciences du Climat et de l'Environnement, LSCE-IPSL (CEA-
CNRS-UVSQ), Université Paris-Saclay, 91191 Gif-sur-Yvette, France.
**now at UniLaSalle - Ecole des Métiers de l'Environnement, Rennes, France.
**Abstract**
Agriculture is the main source of ammonia ($NH_3$) in France, an important gaseous precursor
of atmospheric particulate matter (PM). National and even more global emission inventories
are known to have difficulty representing the large spatial and temporal variability inherent to
atmospheric $NH_3$. In this study, we compare $NH_3$ emissions in France during the spring 2011
from (i) one reference inventory, the TNO inventory, and two alternative inventories that
account in different manners for both the spatial and temporal variabilities of the emissions
(ii) the $NH_3$SAT satellite-derived inventory based on IASI $NH_3$ columns and (iii) the
CADASTRE-CIT inventory that combines $NH_3$ emissions due to nitrogen fertilization
calculated with the mechanistic model VOLT'AIR on the database of the CADASTRE_$NH_3$
framework and other source emissions from the CITEPA. The total spring budgets at the
national level are higher when calculated with both alternative inventories than with the
reference, the difference being more marked with CADASTRE-CIT. $NH_3$SAT and



CADASTRE-CIT inventories both yield to large $NH_3$ emissions due to fertilization on soils
with high pH in the northeastern part of France (65 $ktNH_3$ and 135 $ktNH_3$, respectively, vs 48
$ktNH_3$ for TNO-GEN), while soil properties are not accounted for by the TNO-GEN
methodology. For the other parts of France, the differences are smaller. The timing of
fertilization and associated ammonia emissions is closely related to the nitrogen requirements
and hence the phenological stage of the crops, and therefore tothe crop-year's specific weather
conditions. Maximum emissions are observed in March for 2011 for some regions for both
alternative inventories, while April is the period with maximum emissions for the reference
inventory whatever the region or the year. Comparing the inventories at finer temporal
resolutions, typically at daily scale, large differences are found. The convergence of
alternative, independent and complementary methods on the spatiotemporal representation of
the spring $NH_3$ emissions particularly over areas where the contribution of mineral fertilizer
spreading to the spring budget is strong, encouraging further developments in both prospected
complementary directions, as this will help improving national $NH_3$ emission inventories.

## 48    1. Introduction

France is a major crop producer and a major exporter of agricultural and food products. In
2014, itproduced 2%, 4%, 5%, 8%, 8% and 14% of the global production of maize,
sunflower,wheat, barley, rapeseed and sugar beet, respectively [Food and Agriculture
Organization of the United Nations FAO, Schauberger et al., 2018]. Through this food
cultivation and also due to animal husbandry, agriculture is the main source of ammonia
($NH_3$) in the country. As an important gaseous precursor of particulate pollution, harmful to
human life [Lelieveld et al., 2015; WHO, 2016], ammonia plays an important role in the
regulation of inorganic aerosol concentrations [Erisman and Schaap, 2004, Bauer et al.,
2016], and contributes to N deposition and potential exceedance of critical loads of
ecosystems [Erisman et al., 2007; EEA European Environment Agency, 2014]. In order to
limit air pollution, also responsible for acidification and eutrophication, the new European
National Emission Ceilings Directive 2016/2284, replacing the Directive 2001/81/EC, now
sets ambitious national reduction commitments for ammonia. Ammonia emissions indeed
have to be reduced by 19% in 2030, compared with the 2005 levels [OJEU, 2016].

At the European scale, total $NH_3$ emissions are provided by the European Monitoring and
Evaluation Program (EMEP) [Vestreng, 2005] or by the TNO-MACCIII [Kuenen et al., 2014]



inventories that rely on national annual declarations and estimates of emission factors.
Emissions are accounted for without separating fertilization and livestock. These reference
inventories are widely used by the scientific community to study the impact of pollutant
emissions on the chemical composition of the troposphere and on air quality. Nevertheless,
uncertainties on the quantification of the $NH_3$ emissions are usually estimated to be between
100 and 300% of the annual budgets in the reference inventories [EMEP/EEA, 2016; Kuenen
et al., 2014]. In addition, the temporal and spatial variability may be not well represented in
the reference inventories, as the temporal profiles used do not account for meteorology, soil
properties and other local conditions. Moreover,  fertilizer spreading is of particular interest,
as these are applied during small periods, especially during a few weeks at the end of winter
and early spring. However, the exact timing of fertilizer spreading is difficult to predict, as it
depends on agricultural practices and meteorological conditions, which is not taken into
account in the temporal disaggregation of the reference emission inventories. Both the
inaccurate temporal resolutions and the mis-representation of the spreading emissions largely
explain the difficulty encountered by models to represent seasonal or daily pattern of $NH_3$
concentrations [Menut et al., 2012], and consequently particulate matter levels [Fortems-
Cheiney et al, 2016].
To reduce these uncertainties, a better quantification of agricultural ammonia emissions and
its time and spatial evolution is necessary. In particular, one of the challenges is to capture the
right timing of fertilizer spreading at the weekly or even at the daily scale in order to reflect
the effect of environmental and agronomical conditions on ammonia emissions. To this end,
mechanistic models taking into account meteorological conditions, soil properties and
agricultural practices, have been developed (e.g., for Denmark [Skjøth et al., 2004], for the
UK [Hellsten et al., 2008], and for mineral fertilization in springtime in France [Hamaoui-
Laguel et al., 2014]). Limitations for such approaches come from the fact that detailed
agricultural data needed as input to such models are not available for most of the European
countries. Moreover,agricultural practices of a specific country cannot be extrapolated to
another country [Skjøth et al., 2011].
As an alternative to direct emission modeling, attempts have been made to constrain ammonia
emissions through inverse approaches, based on satellite observed atmospheric ammonia
distributions (e.g., from the Tropospheric Emission Spectrometer TES [Zhu et al., 2013], from
the Infrared Atmospheric Sounding Interferometer (IASI) [Fortems-Cheiney et al., 2016; Van



Damme et al., 2018; Adams et al., 2019] or from the Cross-track Infrared Sounder CrIS
[Adams et al., 2019; Dammers et al., 2019]). In principle, such emission estimates can be
available shortly after observation. The advantage of satellite-derived estimates is also that
these can be derived globally, at a high temporal scale (e.g., daily scale under clear sky). The
downside of these however, is that they do not provide information on the underlying sources
of the emissions (fertilizers vs husbandry), or e.g. the date of fertilization, the type of
fertilizers used, the fertilization rates, etc., that could be important for the regulation of $NH_3$
emissions.

In this context, we compare ammonia emissions in France from inventories using the different
approaches mentioned above: (i) the reference, hereafter called TNO-GEN, is the European
inventory based on the annual budgets provided by the TNO-MACCIII inventory [Kuenen et
al., 2014] and seasonal profiles from GENEMIS [Ebel et al., 1997], (ii) a first alternative
inventory, hereafter called $NH_3SAT$, is based on a top-down approach starting from the IASI
derived $NH_3$ columns ; (iii) the other alternative inventory, hereafter called CADASTRE-CIT,
is based on a bottom-up approach quantifying $NH_3$ emissions due to nitrogen fertilization
combining spatiotemporal data and calculations performed within the CADASTRE_$NH_3$
framework with the mechanistic model VOLT'AIR ([Ramanantenasoa et al., [2018];
Génermont et al., [2018]) completed with livestock and other source emissions from the
French Interprofessional Technical Centre for Studies on Air Pollution (CITEPA). This study
aims at assessing the potential contribution of better spatial and temporal representation of
fertilization-related ammonia emissions to the quality of ammonia emission inventories. The
improvement is assessed in terms of total budget, spatial distribution and timing of the
emissions. The study period, spring 2011 (from March to May 2011), was chosen following
three criteria. Firstly, because at the time of the study, the last French agricultural data were
available from AGRESTE [AGRESTE, 2014] for the agricultural year 2010-2011, allowing
the application of the CADASTRE_$NH_3$ framework for the quantification of the spatio-
temporal distribution of $NH_3$ emissions due to nitrogen fertilization for this crop year
[Ramanantenasoa et al., 2018;Génermont et al., 2018]. Secondly, ammonia emissions are
enhanced during spring in accordance with N crops requirements [Skjøth et al., 2004;
Ramanantenasoa et al., 2018; Génermont et al., 2018]. Finally, unlike autumn and winter
months, the $NH_3$ spring levels are detectable with a better confidence in the IASI satellite
observations [Viatte et al., 2020], allowing the extension of the preliminary work of Fortems-
Cheiney et al. [2016] to deduce $NH_3$ emissions from the IASI satellite instrument.



The three inventories and methods to build them used for this study are presented in Section 2
and the results of the comparison are given and discussed in Section 3.
**2. Inventories**
The three inventories TNO-GEN, NH₃SAT and CADASTRE-CIT compared in this study are
described in Table 1 and in the following sections. It is worth noting that only the
CADASTRE-CIT inventory provides information onthe respective contribution of
fertilization and livestock emissions. The spatial resolutions of the inventories are also shown
in Table 1. The inter-comparison is made at the 0.5° (longitude) x 0.25° (latitude) resolution.
The outputs of the TNO-GEN and the CADASTRE-CIT inventories have consequently been
aggregated.

| Name | Spatial Resolution (latitude x longitude) | Temporal Resolution | Fertilization emissions | Livestock emissions |
|---|---|---|---|---|
| TNO-GEN | 0.125°x0.0625° | Monthly | - | |
| NH₃SAT | 0.5°x0.25° | Daily | - | |
| CADASTRE-CIT | 0.015625° x0.03125° | Hourly | CADASTRE_NH₃ Ramanantenasoa et al., [2018] and Génermont et al., [2018] | |
| | 0.007825°x0.007825° | Daily | | CITEPA national emissions, temporalized according to Skøjth et al., [2011] |

**Table 1**. *Main characteristics of the different compared inventories*.
**2.1. TNO-GEN**
In this study, the TNO-GEN combines the annual budgets provided by the TNO-MACCIII
inventory and the seasonal profiles to deduce the monthly variability of NH₃ emissions. This
inventory is based on official annual emission data submitted by countries to EMEP/CEIP
(European Monitoring and Evaluation Programme/Centre on Emission Inventories and
Projections) for air pollutants. It is the update of the TNO-MACCII inventory [Kuenen et al.,
2014]. It is an inventory at 0.125°x0.0625° resolution providing annual emissions of NH₃



from the agricultural sector, without separating the contributions from fertilization and
livestock. Hereafter, we use the TNO-MACIII emissions of year 2011. The seasonal profile of
these emissions is prescribed in CHIMERE according to the typical national factors provided
by GENEMIS. This seasonal temporal profile used for the temporalization of emissions -the
same one applied to the entire country- leads to a maximum in $NH_3$ emissions systematically
in April over France [Ebel et al., 1997].

**2.2. $NH_3$SAT**
As a first alternative, a mass-balance approach, which is a common method for the
quantification of short-lived species surface fluxes [Palmer et al., 2003; Jaeglé et al., 2004;
Boersma et al., 2008; Lin et al., 2010] was set-up. We used it to deduce $NH_3$ emissions from
differences between $NH_3$ total columns observed by the IASI instrument and simulated by the
CHIMERE regional chemical transport model (CTM) using the TNO-GEN inventory as
inputs data.
2.2.1 The regional CTM CHIMERE
CHIMERE simulates concentrations of gaseous and particulate chemical species [Menut et
al., 2013; Mailler et al., 2017]. For this study, we used the CHIMERE version 2013a. The
horizontal resolution is given as follows: 0.5° × 0.25° over 17°W/40°E–32°N/70°N, including
115 (longitude) x 153 (latitude) grid-cells. The vertical grid contains 17 layers from the
surface to 200 hPa. This model is driven by the European Centre for Medium-Range Weather
Forecasts global meteorological fields [Owens and Hewson, 2018]. Climatological values
from the LMDZ-INCA global model [Szopa et al., 2008] are used to prescribe concentrations
at the lateral and top boundaries and the initial atmospheric composition in the domain. For
inorganic species, aerosol thermodynamic equilibrium is achieved using the ISORROPIA
model [Nenes et al., 1998]. The parameterization of $NH_3$ dry deposition is unidirectional in
CHIMERE.
2.2.2. The IASI observations
We use data from the IASI-A instrument, flying on a low Sun-synchronous polar orbit aboard
Metop satellite since October 2006, with equator crossing times of 09:30 (descending mode)
and 21:30 (ascending mode) local sidereal time (LST) [Clerbaux et al., 2009]. The spatial
resolution of its observations is about 12x12 $km^2$ at nadir. The algorithm used to retrieve $NH_3$
columns from the radiance spectra is described in Van Damme et al., [2017]. Several



improvements have been introduced since the description of Van Damme et al., [2014] and
the version v1 used in our previous study [Fortems-Cheiney et al., 2016]. In this study, we use
the reanalyzed dataset ANNI-NH3-v2.2R, relying on ERA-Interim meteorological input data
from the European Centre for Medium-Range Weather Forecasts (ECMWF) rather than the
operationally provided Eumetsat IASI Level 2 (L2) data used for the standard near-real-time
version [Van Damme et al., 2017]. We only consider land measurements from the morning
overpass, as IASI is more sensitive at this time to the boundary layer, owing to more
favorable thermal conditions [Clarisse et al., 2010, Van Damme et al., 2014].
The IASI total columns are averaged into "super-observations" (average of all IASI data
within the 0.5°×0.25° resolution of CHIMERE). As suggested by Van Damme et al. [2017],
we no longer use weighted averages for this purpose. We performed a sensitivity test by
selecting the IASI pixels for which the retrieval error does not exceed 100%: the results
didnot significantly change, showing the robustness of the IASI $NH_3$ product (not shown).
The resulting monthly means of IASI $NH_3$ columns from March to May 2011 are shown in
Figure 1 (a to c). The spatio-temporal variability -with the highest values over northeastern
France in March, and over northwestern France in April- is confirmed by the IASI 10-year
and by the CrIS 5-year monthly means shown in Viatte et al. [2020]. Note that the potential of
IASI to provide information at high temporal resolution, up to daily scale, can be hampered
by the cloud coverage as only observations with a cloud coverage lower than 10% are
delivered [Van Damme et al., 2017]. To evaluate the impact of this limitation, the number of
IASI super-observations used to calculate these monthly means, which represents the number
of days over a month covered by IASI, is shown in Figure 1 (d to f). On average, more than
half of the month is sampled by IASI during spring, except in May in the northwestern part of
France. The regions showing large IASI $NH_3$ values are consequently well sampled.

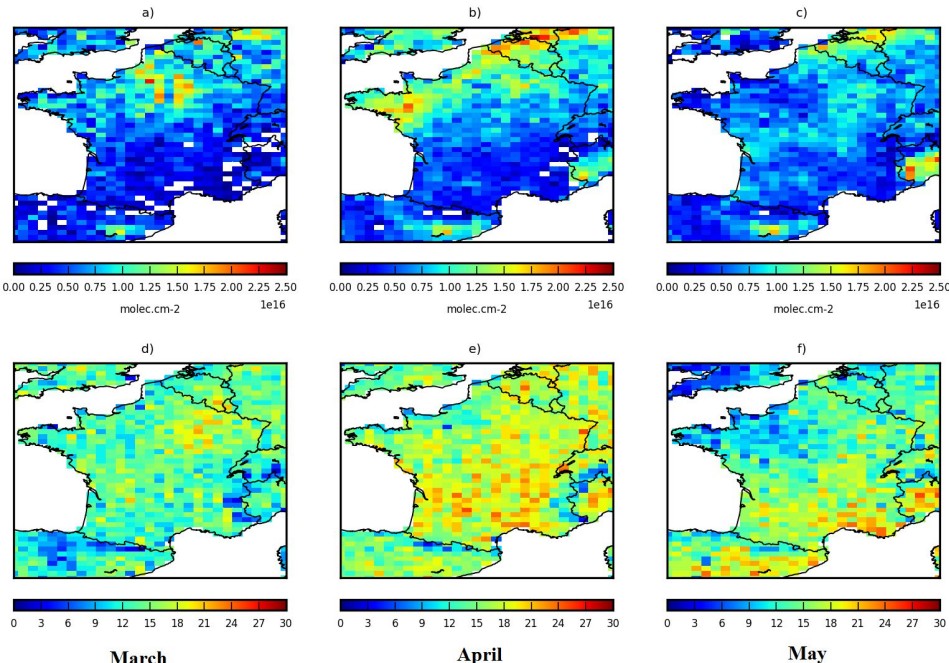

**March**       **April**       **May**


**Figure 1**. (*top) Monthly means of IASI "super-observations" for(a) March 2011, (b) April*
*2011 and (c) May 2011. Units are molec.cm⁻². (bottom) Total number of IASI super-*
*observations permonth in(d) March 2011, (e) April 2011 and (f) May 2011.*

2.2.3. Deducing NH₃SAT emissions

Relative differences between simulated columns by the CHIMERE regional CTM (described
in Section 2.2.1, using the TNO-GEN emissions for the year 2011, described in Section 2.1)
and observed IASI total columns (described in Section 2.2.2) are applied as a corrective factor
to the reference emissions at daily and at grid-cell resolutions over France, from February to
May 2011. As IASI "super-observations" provide one piece of information per day, the
diurnal time profile of reference emissions cannot be improved: we apply the same daily
correction factor to all hourly NH₃ emissions. When IASI is not selected (i.e., observations
with a cloud coverage higher than 10%), the correction is not applied and the emissions
remain equal to the TNO-GEN ones. To compare the emissions with the CADASTRE-CIT
inventory, and their respective simulations with CHIMERE, the correction is only applied
over France here.
With the mass-balance approach, the transport to neighboring cells is assumed negligible
following Palmer et al., [2003]. This approach has been debated by Turner et al. [2012], who
found that non-local sources contribute substantially to columns of short-lived species. Li et
al. [2019] evaluated the ability of both a mass-balance approach and a variational assimilation
to recover known $NH_3$ emissions at different spatial resolutions. At a 2°x2.5° resolution, they
found that both methods yielded similar values. At a 0.25°x0.3125°, the mass-balance
approach led to values about 20% higher compared to the variational ones. With our
0.5°x0.25° resolution, the use of a mass-balance approach would lead to additional errors of
less than about 20% for the quantification of $NH_3$ emissions. This uncertainty is acceptable
and much lower than the uncertainty existing in the annual and national budgets provided by
emission inventories [EMEP//EEA, 2016]. In this context, we choose to perform such mass-
balance approach to deduce $NH_3$ emissions from IASI ANNI-$NH_3$-v2.2R super-observations.
Additional uncertainty comes from the IASI observations. The IASI minimum detection limit
is of about 2-3 ppbv (~4-6.$10^{15}$molecules.$cm^{-2}$) [Clarisse et al., 2010]. The signal-to-noise
ratio therefore presents better performance for regions with high local concentrations (e.g.,
northern part of France, Figure 1) than over low local concentration areas (e.g southern parts
of France, Figure 1). There is no available evaluation for the IASI ANNI-$NH_3$-v2.2R product
used here yet.
**2.3. CADASTRE-CIT**
As a second alternative, a bottom-up approach was set-up based on the finest national
inventories available for anthropogenic sources of ammonia. The CADASTRE_$NH_3$
framework provides such an inventory for organic and mineral fertilization practices. This is
however not the case for the other anthropogenic sources. For livestock emissions, excepted
manure field spreading, the less detailed inventory of the French Interprofessional Technical
Centre for Studies on Air Pollution CITEPA is used. To meet the objectives of better
specialization and temporalization, specific procedures are applied. These inventories are
completed by the TNO-GEN inventory for the emissions of the other sectors.
2.3.1. Fertilization emissions from CADASTRE_$NH_3$
The CADASTRE_$NH_3$ was implemented in order to represent in a realistic way spatio-
temporal variability of French $NH_3$ emissions due to mineral and organic N fertilization, and
is fully described in Ramanantenasoa et al. [2018] and in Génermont et al. [2018]. It has been
constructed through the combined use of two types of resources: the process-based



VOLT'AIR model and geo-referenced and temporally explicit databases for soil properties,
meteorological conditions and N fertilization.
VOLT'AIR is a 1D process-based model predicting $NH_3$ emissions from N fertilizers on bare
soils, from physical, chemical and biological processes [Le Cadre, 2004; Garcia et al., 2012].
It incorporates current knowledge on $NH_3$ volatilization after application of the main types of
organic manure and mineral N fertilizers in the field. It takes into account the major factors
known to influence $NH_3$ volatilization in the field, i.e., soil properties, weather conditions,
cultural practices and properties of mineral fertilizers and organic products. It runs at an
hourly time step at the field scale for a period of several weeks covering thus the entire
volatilization duration of fertilization events.
Local features are attributed to each simulation units, the Small Agricultural Regions (SAR):
local weather conditions (SAFRAN, Météo-France), the dominant soil type of the SAR from
the European Soil Data Center (ESDC), with soil properties provided by the Harmonized
World Soil Database (HWSD) of the Food and Agriculture Organization (FAO); areas
cultivated in crop year 2010-2011 per crop per region derived from the European Land Parcel
Identification System (LPIS, Common Agricultural Policy (CAP) regulations); Nitrogen
fertilization management practices are derived from data ofthe national AGRESTE survey of
cultural practices for arable crops and grassland (Department of Statistics and Forecasting of
the French Ministry of Agriculture) [AGRESTE, 2014]. All input data required by
VOLT'AIR are geographically overlaid and intersected with a Geographical Information
System to generate input combinations in each SAR. Each input combination is used as the
input data for a virtual 300m x 300m field for a simulation using VOLT'AIR. Exact times and
dates of fertilizations are required to run VOLT'AIR, but for the sake of robustness, the
statistical analysis of the survey data has been performed on the base of two-week intervals
for the date of fertilization. Fertilizations are thus randomly distributed within these two-week
intervals in proportion to their respective representation following Ramanantenasoa et al.
[2018]. Each simulation of $NH_3$ emissions is run at an hourly time step for a period of two
months, starting one month before the fertilization in order to calculate soil water content at
the time of application, and ending one month after fertilization, in order to cover the whole
volatilization event.
About 160 000 runs with the VOLT'AIR model have been performed over the crop-year
2010-11to produce ammonia emissionsper hour, per ha, per crop type, per SAR. Emissions



can be aggregated at different spatial and temporal scales. At the spatial scale, they are
weighted with the contribution of (i) each N fertilization management applied to each crop in
each SAR and (ii) the area of the crop cultivated in the SAR. A procedure allows producing
ammonia emissions at the required grid scale for the inventorycomparison: it is based on
cultivated areas for each crop as the key of desegregation-reagregation from the SAR to the
$0.015625° \times 0.03125°$ grid. At the temporal scale, emissions are aggregated over daily, weekly
or monthly bases for the sake of comparison with TNO-GEN and $NH_3SAT$ inventories.
Volatilization taking place over several days, from few days to several weeks, one fertilization
in one field contributes to ammonia emissions over several days or weeks. Weather conditions
effects on overall ammonia emissionsis thus the result of both their effects on fertilization
timing and their effects on volatilization intensity and dynamics over 30 days from fertilizer
application.
As they are not available in the agricultural practice survey, N fertilizations of vegetables,
fruits, and vines are not accounted for in CADASTRE_$NH_3$: their contribution is minor for
France overall, only accounting for 5% of the total agricultural area [AGRESTE, 2010] but
are important in particular regions. As the agricultural practice survey does not provide
information over Corsica, this inventory is completed by the TNO-GEN inventory over this
region (Figure 2).
2.3.2 Livestock emissions
As for the TNO-GEN inventory, French $NH_3$ emissions from livestock for the CADASTRE-
CIT inventory are generated by using annual national emissions provided by CITEPA for
2011 (Figure 3a). Nevertheless, here these emissions have been spatially distributed
differently than for the TNO-GEN inventory. This has been done by using the FAO Gridded
Livestock of the World database with a resolution of 30 seconds of arc. The temporalization
of the emissions has been performed as a function of temperature and wind speed with the
parameterizations of Skøjth et al. [2011] for the different subsectors.
**3. Results and discussion**
First, we analyze the different contributions of livestock and of fertilization to the spring
budget in the CADASTRE-CIT inventory. Then, the comparison of the two alternative
inventories $NH_3SAT$ and CADASTRE-CIT versus the reference inventory TNO-GEN, and
their inter-comparison are made at different temporal and spatial resolutions. We evaluate the
inventories at the national scale and at the scale of the different French administrative



regions(the administrative division in France on level 2 of the unified NUTS territory
classification, NUTS2, shown in Figure 2). We also analyze their spatial variability at the 0.5°
(longitude) x 0.25° (latitude) resolution in order to draw a first picture of the consistency of
the inventories in terms of the spring NH$_3$ total budget and to identify regions of interest.
Finally, we focus on the temporal variability of the identified regions and discuss the
agricultural practices than can influence the variability but also down to which temporal
resolution the comparison of the inventories is relevant.

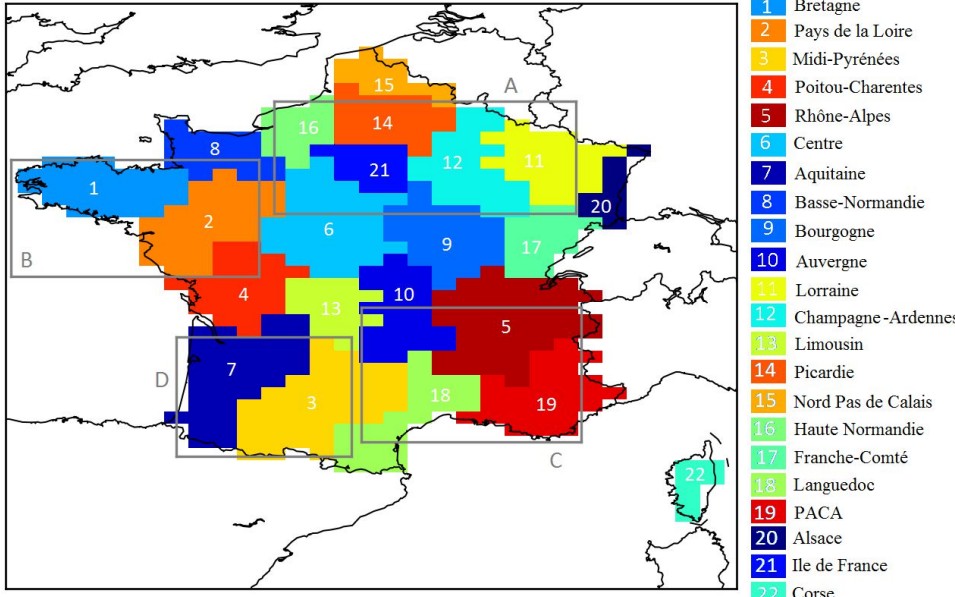


**Figure 2**. *Localization and names of the different French regions as taken into account in this*
*study. The regions are listed according to the TNO-GEN annual budget, in descending order.*
*The grey boxes A, B, C, D describe the domains we respectively call the north-eastern, north-*
*western, south-eastern and south-western parts of France in the following.*

3.1. Respective contributions of different sources
The different contributions of livestock and of fertilization to the annual and to the spring
budget in the CADASTRE-CIT inventory are shown in Figure 3. This figure shows that the
contribution of fertilization to the annual French budget can be strong, with emissions
occurring mainly during spring.


The different contributions of livestock and of fertilization to the spring budget in the
CADASTRE-CIT inventory highlight four different domains of interest. We can see that the
contribution of fertilization on the high emissions of the northeastern part of France (box A) is
strong. For example, the contribution of fertilization is about 99% in the region Champagne-
Ardennes, and about 85% in the region Picardie (Table 2). These emissions are mainly due to
the use of mineral fertilizer over barley, sugar beet, and potato [Ramanantenasoa et al., 2018;
Génermont et al., 2018]. In particular, the use of urea or nitrogen solution and the high soil
pH [Hamaoui-Laguel et al., 2014; Ramanantenasoa et al., 2018; Génermont et al., 2018] -
parameters not taken into account by the TNO-GEN inventory- seem to be the factors
responsible for the high emissions in this domain.
The second domain of interest is the northwestern part of France (box B in Figure 3). Over
this domain with  high emissions, the $NH_3$ emissions are due in roughly equal parts to
livestock (including animal housing, manure storage, and grazing) and to fertilizations, with a
high use of organic manure [Ramanantenasoa et al., 2018]. Livestock farming indeed
produces large amounts of livestock manure available for application on grassland and on
arable crop.
The third domain of interest is the southeastern part of France (box C), showing the smallest
spring $NH_3$ emissions. Finally, the contribution of fertilization on the emissions of the
southwestern part of France (box D) is strong.

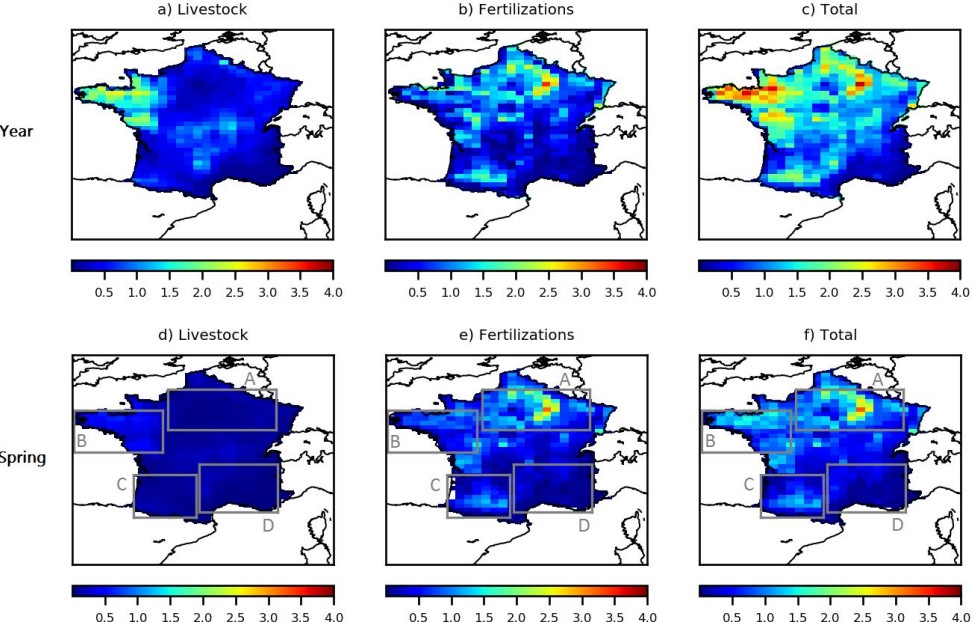


**Figure 3**. *top) Yearly NH₃ emissions due to a) livestock husbandry and manure storage, b) N*

*fertilization (organic and mineral) and c) all sources in the CADASTRE-CIT inventory, in*

*ktNH₃. bottom) the same for spring NH₃ emissions, from March to May 2011.*


### 3.2. French spring NH₃ total budget and its main spatial features

The spring NH₃ total budget is shown in Table 2 at the national scale and at the French

regional scale. The French spring ammonia budgets, calculated for the period from March to

May 2011, estimated by the NH₃SAT (264 ktNH₃) and the CADASTRE-CIT (354 ktNH₃)

inventories are both higher than the TNO-GEN reference one (234 ktNH₃).The CADASTRE-

CIT inventory estimates higher NH₃ spring emissions, by about 30%, than NH₃SAT.

The relative agreement on national budget between TNO-GEN and NH₃SAT must be

nuancedas total budget values from NH₃SAT and TNO-GEN are close but large differences in

the spatial distribution of the French NH₃ emissions between TNO-GEN and both NH₃SAT

and CADASTRE-CIT can be observed (Figure 4).



| | TNO-GEN (in ktNH₃) | NH₃SAT (in ktNH₃) | CADASTRE-CIT (in ktNH₃) | Contribution of the fertilization to the spring budget in CADASTRE-CIT (in %) |
|---|---|---|---|---|
| Regions in the northeastern part of France (box A in Figure 2) | | | | |
| **Champagne-Ardennes** | **8** | **13 (+63%)** | **35 (+337%)** | 99 |
| **Centre** | **13** | **15 (+15%)** | **32 (+146%)** | 81 |
| **Lorraine** | **9** | **12 (+33%)** | **17 (+88%)** | 80 |
| **Picardie** | **8** | **12 (+50%)** | **30 (+275%)** | 85 |
| **Haute-Normandie** | **7** | **9 (+28%)** | **12 (+71%)** | 75 |
| **Ile de France** | **3** | **4 (+33%)** | **9 (+200%)** | 77 |
| Regions in the northwestern part of France (box B in Figure 2) | | | | |
| Bretagne | 34 | 34 (=) | 30 (-12%) | 61 |
| **Pays de la Loire** | **25** | **28 (+12%)** | **29 (+16%)** | 45 |
| Regions in the southeastern part of France (box C in Figure 2) | | | | |
| Rhône-Alpes | 13 | 14 (+8%) | 12 (-8%) | 37 |
| Auvergne | 12 | 12 (=) | 11 (-8%) | 41 |
| Languedoc | 5 | 6 (+20%) | 3 (-40%) | 53 |
| PACA | 5 | 5 (=) | 2 (-60%) | 47 |
| Regions in the southwestern part of France (box D in Figure 2) | | | | |
| Midi-Pyrénées | 18 | 18 (=) | 26 (+44%) | 60 |
| Aquitaine | 12 | 11 (-8%) | 15 (+25%) | 50 |
| Other regions | | | | |
| **Alsace** | **4** | **5 (+25%)** | **7 (+175%)** | 87 |
| **Basse-Normandie** | **11** | **15 (+36%)** | **15 (+36%)** | 68 |
| **Bourgogne** | **11** | **13 (+18%)** | **20 (+81%)** | 55 |
| Franche-Comté | 6 | 6 (=) | 8 (+33%) | 43 |
| Limousin | 8 | 8 (=) | 6 (-25%) | 21 |
| **Nord-Pas-de-Calais** | **8** | **11 (+38%)** | **10 (+25%)** | 97 |
| Poitou-Charentes | 14 | 14 (=) | 25 (+79%) | 63 |
| France | | | | |
| | 234 | 264 (+13%) | 354 (+51%) | 67 |

**Table 2**. *French national and regional budgets of NH₃ spring emissions, from March to May 2011, in ktNH₃. The relative differences compared to the TNO-GEN are presented between brackets, in %. Are marked in bold the regions for which the inventories NH₃SAT and CADASTRE-CIT propose the same sign of relative differences. The contributions of the fertilization emissions to the NH₃ regional spring budget in the CADASTRE-CIT inventory are shown in %.*


Indeed, over France and for the spring budget, the spatial correlation compared to the
CADASTRE-CIT inventory, which should better represent the agricultural practices and their
spatial distribution, is improved when using the $NH_3SAT$ inventory instead of using TNO-
GEN (Pearson correlation coefficient r=0.78 with $NH_3SAT$ against r=0.72 with TNO-GEN).

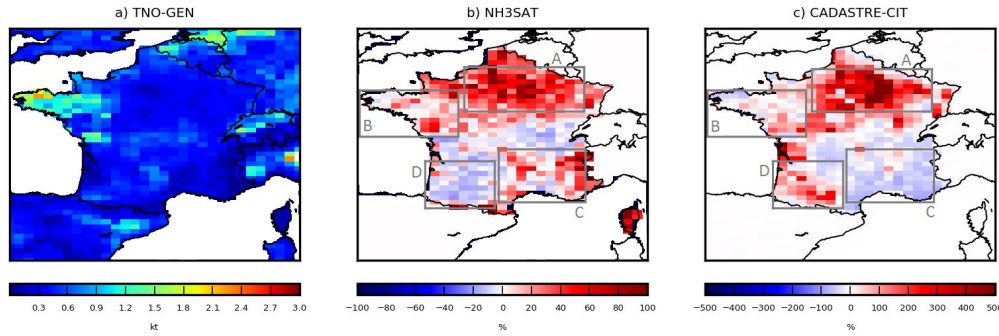


**Figure 4**. *a) French $NH_3$ spring emissions estimated by the TNO-GEN inventory in $ktNH_3$,*
*and relative differences of b) the $NH_3SAT$, and c) the CADASTRE-CIT inventories compared*
*to the TNO-GEN inventory, in %, at the pixel resolution, for the period from March to May*
*2011. Note that the scale is different between 4b) and 4c).*
The northeastern part of France presents the largest difference with the TNO-GEN inventory
(48 $ktNH_3$) for both $NH_3SAT$ and CADASTRE-CIT inventories (65 and 135 $ktNH_3$,
respectively). Indeed, emissions are higher for both inventories compared to TNO-GEN over
the Champagne-Ardennes (+337% and +63%, respectively for CADASTRE-CIT and
$NH_3SAT$, Table 3), Picardie (+275%, and +50%, respectively), Centre (+146% and +15%,
respectively), Haute-Normandie (+71% and +28%, respectively), Lorraine (+88%, and +33%,
respectively) and Ile de France regions (+200%, and +33%, respectively).
The northwestern part of France presents the largest $NH_3$ emissions according to the TNO-
GEN inventory (Figure 4a). The TNO-GEN, $NH_3SAT$ and CADASTRE-CITinventories lead
to similar spring budget (68, 73 and 71 $ktNH_3$, respectively, over this domain.
Over the southeastern part of France, the TNO-GEN and $NH_3SAT$ inventories are also in
quite good agreement in terms of budget (35 and 37 $ktNH_3$, respectively, Table 2) but not in
terms of spatial distribution (box C in Figure 4). On the contrary, CADASTRE-CIT is about
23% lower than TNO-GEN and $NH_3SAT$ (Table 2). One hypothesis to explain the lower $NH_3$
emissions in CADASTRE-CIT is that market gardening is important in this area and not taken



into account in the CADASTRE-CIT inventory [Ramanantenasoa et al., 2018; Génermont et
al., 2018].
Finally, over the southwestern part of France (box D in Figure 4), IASI observations only
trigger slight corrections to the TNO-GEN inventory over this area (29 and 30 ktNH$_3$,
respectively) and CADASTRE-CIT is 36% higher than TNO-GEN and NH$_3$SAT (Table 2).

3.3. Temporal variability of the NH$_3$ emissions at the sub-seasonal scale

Monthly regional budgets have been calculated for the three inventories. Figure 5 presents the
monthly variability of the NH$_3$ emissions from February to May 2011 for the four domains of
interest presented above. February is only displayed hereas a baseline to show the sharp peak
of NH$_3$ emissions in March over some domains.The contribution of the emissions due to
livestock in the CADASTRE-CIT spring budget is also given.

The TNO-GEN inventory shows rather similar NH$_3$ emissions from March to May for all
regions, with a slight maximum in April (Figure 5a), imposed by the used GENEMIS monthly
profiles for the temporalization of emissions [Ebel et al., 1997]. On the contrary, over the
northeastern part of France, both NH$_3$SAT and CADASTRE-CIT inventories show a
maximum in March, and a decrease until May by about a factor of 1.5 to 2 (Figure 5a). We
calculated the monthly contribution of livestock to the NH$_3$ emissions based on CADASTRE-
CIT, which allows one to separate this contribution from the fertilization one. As in Figure 3,
Figure 5a confirms that NH$_3$ emissions are mainly due to fertilization in the northeastern part
of France, and shows that the seasonal variation is mainly driven by this contribution,which
confirms the hypothesis formulated in introduction. CADASTRE-CIT shows larger values
than NH$_3$SAT which might be partly due to a possible low bias in the IASI observations
[Dammers et al., 2017] combined to the fact that the TNO-GEN inventory (negatively biased
compared to CADASTRE-CIT) is used as a priori for the mass balance approach with no
correction applied to the a priori when IASI observations are not available.
To go further in the comparison, we analyzed the daily variability of the NH$_3$SATand
CADASTRE-CIT inventories -TNO-GEN representing no daily variability (Figure 6). To
interpret the results, some limitations of the inventories have to be considered. For NH$_3$SAT,
the corrections applied to the TNO-GEN emissions are only applied for clear-sky conditions
when IASI observations are available. In the CADASTRE-CIT, fertilization days are
randomly selected within two-week intervals of application extracted from the farm survey


analysis [Ramanantenasoa et al., 2018], thus the actual day of fertilization is unknown.
However, the $NH_3$ volatilization is continuous over several days after spreading reducing the
uncertainty introduced by this random selection. Moreover, the random selection is made at
the field scale (see section 2.3), then spatially averaging at the CHIMERE resolution should
also smoothed the random selection effect. Both effects of weather conditions on fertilization
timing, on the one hand, and on volatilization intensity and dynamics at the time of
application and after application, on the other hand, are realistically produced. Hence, the $NH_3$
volatilization is continuous over several days after spreading introducing additional
smoothing.
The CADASTRE-CIT inventories presents a high day-to-day variability from March to May
2011 (Figure 6a) with several strong maxima of emissions - characteristic of emissions due to
fertilizer application and a significant effect of the varying meteorological conditions. Also
for $NH_3SAT$, day-to-day variability is large. However, $NH_3SAT$ and CADASTRE-CIT
maxima are not very well correlated. Over the first 15-days in March 2011, the high
emissions occurring from the 9th to 16th in CADASTRE-CIT are not reproduced by the
$NH_3SAT$ inventory, potentially because of a lack of IASI coverage for this period (only about
40% of the domain –Figure 6a). $NH_3SAT$ shows an emission maximum one week earlier
from the 1st to 7th. This time gap in emission maxima could be explained by the random
selection of fertilization days in CADASTRE-CIT. Over the last 15 days in March 2011 and
over the first 15 days in April 2011, the maxima of emissions estimated by $NH_3SAT$ and
CADASTRE-CIT are more correlated, (e.g., from 7th to 11th April in CADASTRE-CIT
versusfrom 9th to 11th in $NH_3SAT$). Finally, CADASTRE-CIT still shows high emissions in
the last 15 days of April and in May 2011, particularly related to the use of fertilizer over
corn. Despite a good coverage with IASI observations, no specific high emissions are derived
from IASI for the same periods.
Over the northwestern part of France, the CADASTRE-CIT inventory is in agreement with
TNO-GEN, with a slight maximum in April (Figure 5b). The $NH_3SAT$ inventory shows a
maximum in March, relatedto the high emissions seen during the first week of March 2011
(Figure 6b). However, the monthly differences are much smaller than for the northeastern part
of France. Also for this domain, the day-to-day variabilities provided by CADASTRE-CIT
and $NH_3SAT$ are mostly uncorrelated: a strong sharp maximum of emissions during the first
week of March 2011 is seen in $NH_3SAT$ -but not reproduced in CADASTRE-CIT (Figure



6b). The highest daily emissions in CADASTRE-CIT occur during the two last weeks of
April 2011 and are not reproduced by NH$_3$SAT.

Over the southeastern and southwestern parts of France, month-to-month variations and
emission amounts are much smaller than for the two previous domains(Figures5-6c and
Figures5-6d, respectively). In the southeastern part of France, NH$_3$SAT emissions are slightly
larger than TNO-GEN and CADASTRE-CIT at the beginning of the period (February and
March, Figure 5c). The fertilization contribution to CADASTRE-CIT emissions slightly
decreases at the end of the period. In the southwestern part of France, NH$_3$SAT and TNO-
GEN are very similar (Figure 5d) and CADASTRE-CIT is slightly larger at the end of spring.
This increase is mainly related to fertilization emissions, the livestock contribution being
stable. Again, daily time series between both NH$_3$SAT and CADASTRE-CIT inventories are
uncorrelated (Figures 6c, d).




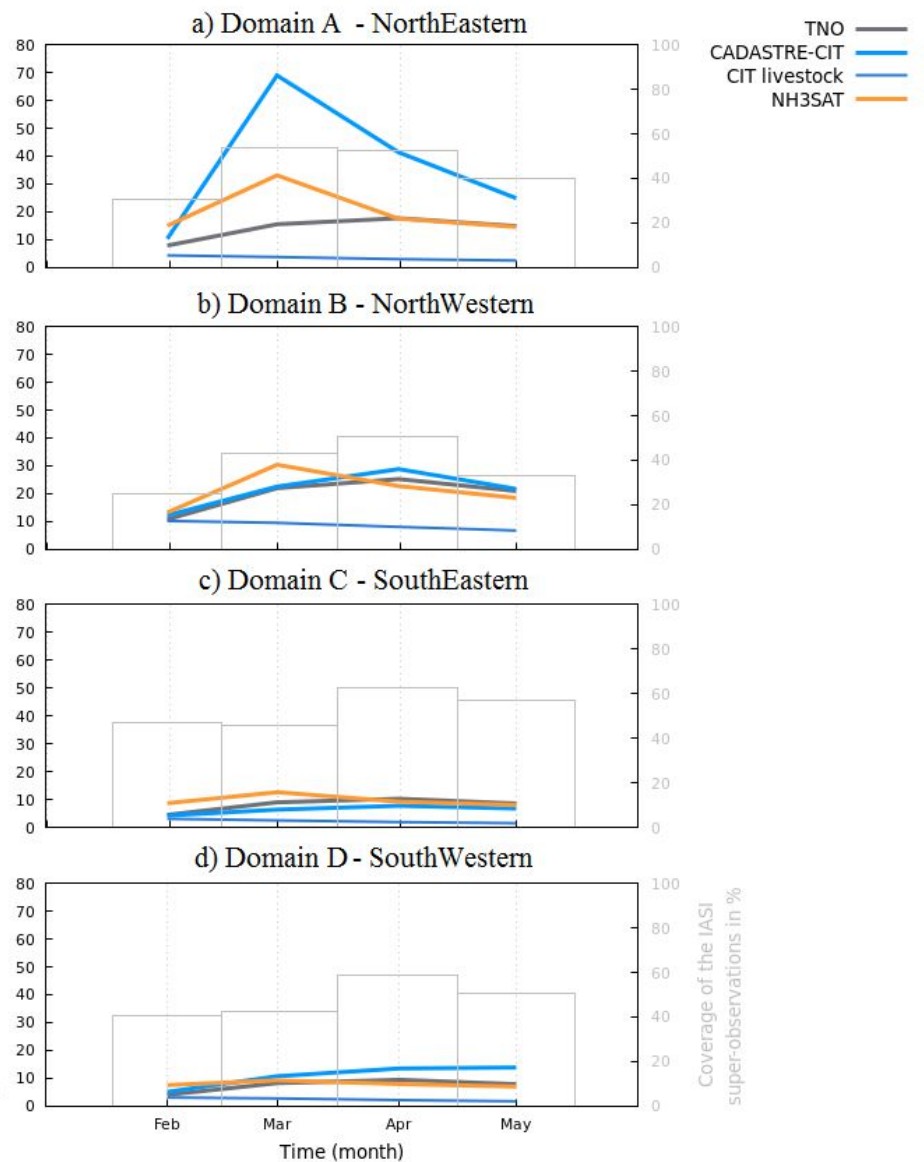

**Figure 5**. *Time series of monthly NH₃ emissionsestimated by TNO-GEN (in grey), by NH₃SAT (in orange), and by CADASTRE-CIT (in blue) inventories, from February to May 2011, for (a) domain A (north-eastern), (b) domain B (north-western), (c) domain C (south-eastern), and (d) domain D (south-western), as defined in Figure 2. The contribution of the emissions due to livestock in the CADASTRE-CIT monthly budgets is also given. Units are ktNH₃/month. The monthly regional coverage of the IASI super-observations is given in % (in grey).*

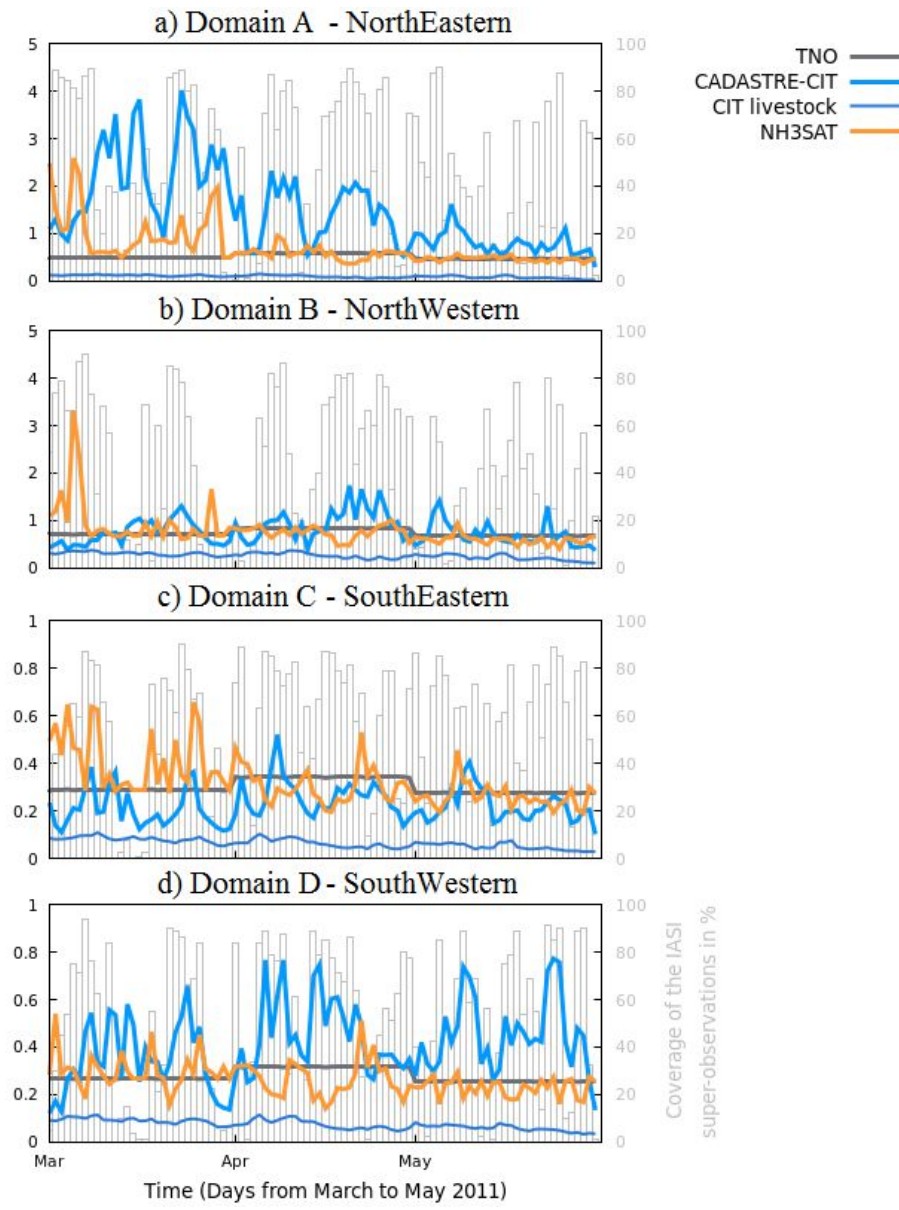

**Figure 6**. *Time series of daily NH₃ emissions estimated by TNO-GEN (in grey), by NH₃SAT*

*(in orange), and by CADASTRE-CIT (in blue) inventories, from March to May2011, for (a)*

*domain A (north-eastern), (b) domain B (north-western),( c) domain C (south-eastern), and*

*(d) domain D (south-western), as defined in Figure 2. Note that the scale is different between*

*a),b) and c),d). Units are ktNH₃/day. The contribution of the emissions due to livestock in the*





*CADASTRE-CIT daily variability is also given. The daily regional coverage of the IASI super-*
*observations is given in % (in grey).*
**Conclusion**
In this study, we performed an inter-comparison of two alternative inventories with the TNO-
GEN reference inventory that quantify the French $NH_3$ emissions during spring 2011. One of
the main conclusion of this study is that over regions with large mineral fertilizer use, like
over North-Eastern France, induced $NH_3$ emissions are probably considerably underestimated
by the TNO-GEN reference inventory, as both the $NH_3SAT$ (constrained by IASI
observations) and CADASTRE-CIT (process level oriented), show much larger values. For
instance, over northeastern France, $NH_3SAT$ and CADASTRE-CIT show respectively a factor
1.4 and 2.8 larger spring 2011 emissions than TNO-GEN. Over the whole France, $NH_3$
emissions are still more than 50% larger in CADASTRE-CIT than in TNO-GEN. Average
French $NH_3SAT$ emissions are about 10% larger than TNO-GEN ones. Over the southern part
of France, with a more diverse agriculture as compared to the crop intensive one in North-
eastern France, differences between the inventories are on the whole lower, and signs between
CADASTRE-CIT / TNO-GEN and $NH_3SAT$ / TNO-GEN corrections are often opposite for
different regions.

Month-to-month variations are again much more pronounced over North-Eastern France and
show a maximum in March for both CADASTRE-CIT and $NH_3SAT$. Day-to-day variations
are large in CADASTRE-CIT and $NH_3SAT$, roughly a factor of 5 between minimal and
maximal values. This shows the interest in evaluating $NH_3$ emissions at a daily scale because
this input is required for chemistry transport modeling of particulate matter formation and
thus air quality. However, time-series delivered by CADASTRE-CIT and $NH_3SAT$ are
uncorrelated for all considered regions. This result can be partly explained by the lack in IASI
$NH_3$ column observations under partially cloudy conditions, and by the fact that available
information on agricultural practices is resolved at a two weeks scale.

Thus, as a general conclusion, use of observational constrained or process oriented emission
inventories is clearly of added-value for estimating better monthly averages over French areas
with intensive mineral fertilizer use, but capacity for delivering day-to-day variations is not
yet proven. This warrants further studies, both refining hypotheses on days chosen by farmers
for fertilizer spread out as a function of meteorological conditions, and, acquiring and using





continuous surface NH$_3$ measurements for validating satellite or process derived NH$_3$
emission variability.

**Competing interests**

The authors declare that they have no conflict of interest.

**Author Contribution**

All authors have contributed to the manuscript writing (main authors A.F-C, G.D, S.G and
MB). A.F-C has performed the mass-balance approach to deduce NH$_3$ emissions from NH$_3$
total columns observed by the IASI satellite instrument. K.D, J-M.G and S.G have performed
the bottom-up approach providing the CADASTRE_NH$_3$ inventory for organic and mineral
fertilization practices. F.C has compiled this CADASTRE_NH$_3$ inventory with livestock
emissions from the CITEPA. L.C, P-F.C, M.V.D and C.C are the PIs of the IASI NH$_3$
product. All authors discussed the results and contributed to the final paper.

**Code and Data Availability**

The CHIMERE code is available here: www.lmd.polytechnique.fr/chimere/.
The IASI ANNI-NH3-v2.2R data are freely available through the AERIS database:
https://iasi.aeris-data.fr/nh3/.

**Acknowledgements**

For this study, A. Fortems-Cheiney and K. Dufossé were funded respectively by the Amp'Air
and the PolQA Primequal projects, under agreement numbersADEME 1660C0013 and
1662C0023, respectively. L.C and M.V.D are respectively research associate and postdoctoral
researcher with the Belgian F.R.S-FNRS. The authors are indebted to all those who provided
input data for CADASTRE_NH$_3$ 2010-11 crop year: the *Service de la Statistique et de la*
*Prospective* (SSP), Department of Statistics and Forecasting of the French Ministry of
Agriculture, for the French cultural practices survey, supported by a public grant overseen by
the French National Research Agency (ANR) as part of the "Investissements d'avenir"
program (reference: ANR-10-EQPX-17 – *Centre d'Accès Sécurisé aux Données* – CASD);
Météo-France for the data on weather conditions; the *Agence de Service et de Paiement* (ASP)
and the *Observatoire du Développement Rural* (ODR) service unit for data on Land Parcel
Identification System (LPIS) for France. The authors are also grateful to
M.M.J. Ramanantenasoa for her technical assistance, funded by the Cadastre_NH$_3$ project
under agreement number ADEME 1081C0031. IASI is a joint mission of EUMETSAT and
the Centre National d'Etudes Spatiales (CNES, France). The authors acknowledge the AERIS



data infrastructure for providing access to the IASI data in this study.This research has been supported by the Belgian State Federal Office for Scientific, Technical and Cultural Affairs (Prodex arrangement IASI.FLOW). This work was granted access to the HPC resources of TGCC under the allocation A0050107232 made by GENCI.

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
