# Peer review of "ammonia emissions in France?"

_Atmospheric Chemistry and Physics, 2020_

## Referee Comment (RC1) · Anonymous Referee #1 · 15 Jun 2020

Referee omment on "Do alternative inventories converge on the spatiotemporal representation of spring ammonia emissions in France?" by Audrey Fortems-Cheiney et al.

GENERAL COMMENTS

The manuscript compares 3 inventories of NH3 over France for the year 2011. The inventories are referred to as the "TNO" dataset (from a European inventory, based on reported national totals), the "NH3SAT" dataset (based on IASI inventories), and the "CADASTRE" dataset (based on a highly detailed model). The emission inventories are compared to each other, and used in simulations with a regional transport model

from which simulated concentrations are compared too. Since NH3 is an important precursor for aerosols and nitrogen deposition in agriculture intensive regions, this study provides a useful contribution to air quality modelling. I would therefore recommend to publish the manuscript after some minor clarifications.

The results show that to a large extend the three inventories are in agreement with each other, with exception of northern France during the spring period. An important conclusion is that application of mineral fertilizer in combination with certain soil properties could be very relevant when compiling NH3 emission database. When this is taken into account, as is done in the "CADASTRE" set, then simulated NH3 concentration columns are in better agreement with IASI observations than simulations driven by the "TNO" emissions, which uses a more simple approach for spatial distribution of emissions.

It would be useful of the authors could provide an outlook on how their results should be used in future. It is clear that a more detailed emission model as used in "CADASTRE" could provide better emission inventories, but from the provided information it seems not possible to apply this over, for example, the rest of Europe. In addition, it seems that in spite of its high detail also "CADASTRE" is not able predict the timing of emission right. But could the information that is used in "CADASTRE" find it's way into the "official" inventories such as that from "TNO" ? For example, would it be sufficient to just have maps of fertilizer use and soil properties for a better spatial redistribution, which is now primary based on live-stock densities? Or is modelling of emissions to uncertain anyway, and should we rely most on (satellite) observations? Also, should the official national reporting of NH3 emissions be changed following the results of this study? Some clear recommendations on this would be useful.

SPECIFIC COMMENTS

Table 1. Some clarifications on the temporal resolutions would be useful. Emission inventories like "TNO-GEN" are usually accompanied with profiles for month-of-the-year

(as used here), but also "day-of-the-week" and "hour-of-the-day". That would change the resolution to "hourly", although the uncertainty is high of course. For "NH3SAT" there is no full daily resolutions, since data coverage in time is not 100% as shown in Figure 1 lower panel. For "CADASTRE" line 525-526 mentions two-weekly data on agricultural practices; what does this do with the "daily" temporal resolution of the emission model?

Line 154. CHIMERE has not been mentioned before, maybe point forward to section 2.2.1.

Paragraph 192-196. It is not clear to me how many IASI pixels are typically used for a single "super-observation". The "robustness" of the data is mentioned, does that mean the the variability between nearby pixels is low? Figure 1 shows some some strong gradients however. Or thus "robustness" more refer to the temporal and regional differences?

Line 219-221. When no IASI super-obs is available, the "NH3SAT" inventory uses the "TNO-GEN" data. Given Figure 1 that would mean that for at least half of the days in a month the "TNO" emissions are used. Given the regional differences between the "TNO" and "NH3SAT" inventories, wouldn't it be logical to assume some kind of persistency here? Thus, use the latest "NH3SAT" value if for some day there is no super-obs available? Also temporal interpolation would be an option. How would that change the emission totals? And the conclusions on temporal variability, e.g. at line 433?

Figure 4. This figure would be better interpreted if also maps with absolute emissions for "NH3SAT" and "CADASTRE" are added, and not only the differences. In general, the comparisons are now often "against" the "TNO" set, while I would think that it should not serve as a "truth" but just as one of the 3 inventories, each with their advantages and disadvantages.

TECHNICAL CORRECTIONS

Text changes needed?

45. "encourage" instead of "encouraging"

72. "may not be well"

85. "... its temporal and spatial ..."

129. just "crops", not "N crops"

219. "When IASI is not available ..."

240. "There is no evaluation available ..." ?

246. "excluding" instead of "except"?

296. "Weather condition effects"

325. "that can"

379. "Regions are marked in bold ..."

440. "after spreading, reducing"

448. "inventory"

Figure 5 and 6: I would call this color "black" rather than "gray" . . .

At many places it seems that white-space is missing (or is this something in the pdf?):

38. "to the"

50. "it produced"

51. "sunflower, wheat"

93. "Moreover, agricultural"

139. "on the"

196. "did not"

272. "of the"

291. "inventory comparison"

297. "emissions is"

320. "regions (the ...)"

371. "nuanced as"

433. "NH3SAT and"

446. "related to"

475. "domains (Figures ..."

---

## Referee Comment (RC2) · Anonymous Referee #2 · 26 Jun 2020

Summary

The authors present an inter-comparison of two alternative inventories with the European reference inventory (TNO) to quantify the French NH3 emissions during spring 2011; (i) NH3SAT inventory which is based on a top-down approach of correcting TNO NH3 emissions based on total column observations and (ii) CADASTRE-CIT inventory which is built from the bottom-up based on modeled NH3 emissions related to fertilizer application and animal husbandry. There is a thorough comparison of inventories based on different regions of France with known anthropogenic NH3 emitting activities. The work specifies which regions in France, represented in the European reference
inventory, need to be improved upon and highlights the general conclusion of improving NH3 emissions inventories based on measurements and process knowledge is required.

Inventories are key inputs in forecasting air quality, so the goal of this study to build a more representative inventory over France is important work. Building NH3 inventories is very challenging because of its varied lifetime in the atmosphere and complex exchange mechanisms making its spatial distribution and temporal behavior difficult to predict. Atmospheric NH3 is an important precursor to PM and can also damage N-sensitive ecosystems, therefore, refined emissions inventories are needed for air quality modeling and monitoring emissions reductions. This study details methods to refine inventories, therefore, I would recommend publishing this manuscript after some revisions.

Major Comments

There have been other global inventories built by the atmospheric community using similar methods that have not been mentioned and would add to the discussion of the authors' work.

Work from Zhang et al. (ACP, 18, 339-355, 2018), who reconcile bottom-up and top-down inventories, also show including more detailed information on crop-specific fertilizer application practices and met factors does a better job at reproducing spatial and seasonal variations in China, which seems to be similar in this study, but in France.

How do the two alternative inventories compare to currently available inventories, aside from TNO? How does CADASTRA-CIT compare with the MASAGE_NH3 inventory? and other global inventories that represent France, such as EDGAR? Is TNO built upon any of these inventories already?

Atmospheric NH3 is known to undergo bi-directional exchange with surfaces and this aspect is not discussed. The NH3SAT is generated based on chemical transport modeling of TNO, however, the CHIMERE model parameterizes NH3 dry deposition unidi-rectionally. Is this a limit of the model? If NH3 dry deposition is assumed to be a net sink, then in most cases that would underestimate atmospheric concentrations. How does this impact NH3 estimates? and to what degree?

Can the authors comment on how well CHIMERE can predict particulate NH4+? How much does that influence NH3 estimates?

What are the limits of the Volt'air model? It is usually used to predict emissions from slurry applications, so it doesn't account for a crop canopy. Does that matter? Are there any fast-growing crops that would sprout in the first month in which the model was run?

What is the availability of ground-based NH3 measurements in regions which have the most variation? If so, how do they compare with IASI observations if we assume the total column of NH3 is all at the surface? Were ground-based NH3 concentrations used as an a priori for IASI total column calculations?

Minor Edits

Line 401 is missing a parenthesis - (68, 73, and 71 ktNH3, respectively).

I am not sure if there is something funky with the text editor file that messes with the spacing in the pdf, but there are some words that have been compounded throughout the article.

IASI reference should also include Clarisse, L., Clerbaux, C., Dentener, F., Hurtmans, D., and Coheur, P.: Global ammonia distribution derived from infrared satellite obser-vations, Nat. Geosci., 2, 479–483, 2009.

---

## Referee Comment (RC3) · Anonymous Referee #3 · 1 Jul 2020

GENERAL COMMENTS

The manuscript submitted by Fortems-Cheiney et al. assesses different methods to determine NH3 emissions that are used in the scientific community. The authors compare three state-of-the art approaches for NH3 emission retrievals and discuss their spatial and temporal differences over a period in Spring 2011, which also includes the time of fertilizer application. While the analysis was performed for France, the discussed discrepancies between methods is also of high interest to researcher focusing on other regions. Due to the general high uncertainties in NH3 emission estimates, the presented study is of high relevance for improving chemical transport models.

The manuscripts is well and clearly written and the analysis and interpretation of results is sound. Given the relevance of the topic for air quality, nitrogen deposition and atmospheric chemistry in general, I suggest approving the manuscript for publication in Atmospheric Chemistry and Physics after addressing the following comments:

SPECIFIC COMMENTS

L. 34-35: I suggest to clarify that the given values are the cumulative emissions from March to May. It is mentioned above, however, it would be good to restate here to not confuse it with an emissions flux.

L. 123: I assume the study period was from 1 March to 31 May? Still, the day should be included in the period description.

L. 217-219: Above it is stated that only morning overpass measurements are used. This may introduce a systematic bias to the NH3SAT emissions as emissions may be larger during midday due to higher temperatures or increased activity (e.g. mobile emissions). If I understand it correctly, the effect also depends on the TNO-GEN input data for CHIMERE, i.e. if monthly or a diurnal profile is applied to that. While I agree with the decision to only use morning overpass data, this potential bias should be explained and how it may impact the findings (e.g. could it explain why NH3SAT emissions are lower than those of CADASTRE-CIT in northeast, where fertilizer emission are high?).

L. 219-221: Could one also take only those periods where the IASI quality criteria are met and omit the other periods for the analysis instead of using the TNO-GEN emissions? How would that change the average correction factors presented in this study?

L. 246-247: From this paragraph alone it is not clear whether the manure field spreading is included the organic fertilizer application, which I would assume. If that is the case, you may add here ". . ., which is part of the organic fertilizer emissions" or similar.

L. 314: In the Results & Discussions section the authors describe the differences between the different approaches and also give here and there some suggestions why these differences occur. Still the discussions can be extended at the end of the section, e.g. by elaborating more on the uncertainties of each approach, leading to recommendations for future research directions and at which part each of the approaches should be improved (e.g. need for implementation of bi-directional exchange module in CHIMERE?). Although the paper is targeted on France, the findings are also insightful for users of NH3 emission estimates in other parts of the world. Therefore, relating the findings in the discussion also to inventories/approaches used in other regions would in my opinion make the manuscript stronger and attract more interest by a wider community.

L. 428: I suggest restating here in short the hypothesis from the introduction.

L. 405-408: Would market gardening be included in the TNO-GEN inventory? If not, an underestimation in that region would also apply to the TNO-GEN inventory.

TECHNICAL COMMENTS

Some words are accidentally merged, for example in lines 38, 50, 51, 93, 139, 196, 197, 207, 209, 210, 211, 272, 287, 291, 320, 371, 381, 400, 433, 466, 475 and 478.

The description and spelling of regions like "North-Eastern France" vs. "northeastern France" should be consistent throughout the manuscript.

L. 26: The sentence structure can be improved as it can be misleading what the actual methods are which the authors refer to. For example, I suggest inserting ":", "namely"... or something similar after "emissions".

L. 292: I am not familiar with this terminology but there might be better terms than "desegregation" and "reagregation" in English.

L. 379: I suggest using "In bold are marked...".

L. 401: A closing parenthesis is missing.

---

## Author Comment (AC1) · 7 Sep 2020

**Reviewer #1**

We wish to thank the referee for his/her helpful comments. The full reviews are copied hereafter and our responses are inserted. The comments of the reviewer are in normal black and our answers in bold.

**GENERAL COMMENTS**

The manuscript compares 3 inventories of NH3 over France for the year 2011. The inventories are referred to as the "TNO" dataset (from a European inventory, based on reported national totals), the "NH3SAT" dataset (based on IASI inventories), and the "CADASTRE" dataset (based on a highly detailed model). The emission inventories are compared to each other, and used in simulations with a regional transport model from which simulated concentrations are compared too. Since NH3 is an important pre-cursor for aerosols and nitrogen deposition in agriculture intensive regions, this study provides a useful contribution to air quality modelling. I would therefore recommend to publish the manuscript after some minor clarifications.

The results show that to a large extend the three inventories are in agreement with each other, with exception of northern France during the spring period. An important conclusion is that application of mineral fertilizer in combination with certain soil properties could be very relevant when compiling NH3 emission database. When this is taken into account, as is done in the "CADASTRE" set, then simulated NH3 concentration columns are in better agreement with IASI observations than simulations driven by the "TNO" emissions, which uses a more simple approach for spatial distribution of emissions. It would be useful of the authors could provide an outlook on how their results should be used in future.

Thank you for this suggestion, which allows highlighting the implications of our study. We added a couple of sentences at the end of the Conclusion section:

"Yet, current results of our study have important implications for air quality modelling over Europe. The important changes in the spatial distribution of NH3 emissions as a function of soil properties are of general concern not only for France, but for whole Europe. Soils are alkaline or neutral (pH>6) not only over North-Eastern France, but also over large parts of Italy, eastern Spain, or eastern Germany [Reuter, 2008]. Over these regions, our study suggests potentially larger NH3 emissions than with a constant emission factor treatment, with impacts then on fine particle formation. These features should be included in "operational" emission inventories used for air quality modelling."

**Reuter, H.I., Lado, L.R., Hengl, T. and Montanarella, L.: Continental-scale digital soil mapping using European soil profile data: soil pH, Hamburger Beiträge zur Physischen Geographie und Landschaftsökologie – 92 Heft 19/2008, pp. 91-102, 2008.**

It is clear that a more detailed emission model as used in "CADASTRE" could provide better emission inventories, but from the provided information it seems not possible to apply this over, for example, the rest of Europe.

Indeed, Skjøth et al., [2011] already showed that agricultural practices of a specific country cannot be extrapolated to another country, right now. Specific adaptations to national databases (especially agricultural practices) should be performed beforehand. Also, soil properties need to be updated for each country. However, we think it is worthwhile to undertake such an effort in future work.

In addition, it seems that in spite of its high detail also "CADASTRE" is not able predict the timing of emission right. But could the information that is used in "CADASTRE" find it's way into the "official" inventories such as that from "TNO"? For example, would it be sufficient to just have maps of fertilizer use and soil properties for a better spatial redistribution, which is now primary based on livestock densities? Or is modelling of emissions too uncertain anyway, and should we rely most on (satellite) observations?

**Yes indeed, while predicting day to day timing of NH3 emissions is still in research area, modulation of emission factor (per fertilizer use) taking into account soil properties is a realistic perspective for official emission inventories. This is what we put into the perspectives added to the manuscript (just above).**

**SPECIFIC COMMENTS**

Table 1. Some clarifications on the temporal resolutions would be useful. Emission inventories like "TNO-GEN" are usually accompanied with profiles for month-of-the-year (as used here), but also "day-of-the-week" and "hour-of-the-day". That would change the resolution to "hourly", although the uncertainty is high of course. For "NH3SAT" there is no full daily resolutions, since data coverage in time is not 100% as shown in Figure 1 lower panel. For "CADASTRE" line 525-526 mentions two-weekly data on agricultural practices; what does this do with the "daily" temporal resolution of the emission model?

Indeed, we agree. The legend of Table 1 has been changed in « Main characteristics of the different compared inventories before their aggregation/disaggregation for the intercomparison." Clarifications on the temporal resolutions about TNO have also been added in Table 1 and in Section 2.1: "The emissions remain constant between days in each month and between hours in each day."

Indeed, for CADASTRE\_NH3, available information on agricultural practices is resolved at a two weeks scale. It is linked to the data source and treatment. In fact, the dates of fertilization are extracted from a national survey on the basis of a relatively small number of farms per region and per crop: they are then exploited through statistical analyses. In order for the extracted results to be meaningful, we have chosen this fortnightly basis, which is also consistent with the surveys of previous years (fortnightly basis). The dates for which simulations are carried out are randomly selected from the fortnightly surveys in proportion to their representation: this method does not lead to an exhaustive choice of dates. A sensitivity analysis of the dynamics and extent of volatilization is currently being carried out to evaluate the possibility of changing the number of draws in relation to the quality of the outputs in view of the dynamics and extent of volatilization. Results of these tests and potential evolution of the Cadastre\_NH3 framework would be published at a later date.

Line 154. CHIMERE has not been mentioned before, maybe point forward to section 2.2.1. **Indeed, the reference to CHIMERE in this sentence has been removed.**

Paragraph 192-196. It is not clear to me how many IASI pixels are typically used for a single "super-observation".

1, 2 or 3 IASI pixels are typically used for a single « super-observation ». We have completed the definition of the « super-observation »: « average of all IASI data within the  $0.5^{\circ} \times 0.25^{\circ}$  resolution and for the given CHIMERE physical time-step of about 5/10 minutes)"

The "robustness" of the data is mentioned, does that mean the variability between nearby pixels is low? Figure 1 shows some strong gradients however. Or thus "robustness" more refer to the temporal and regional differences?

**Indeed, the robustness here refers to the temporal and regional differences. We have completed the sentence: « the results about the temporal and spatial variability of the NH3 French emissions presented in Section 3".**

Line 219-221. When no IASI super-obs is available, the "NH3SAT" inventory uses the "TNO-GEN" data. Given Figure 1 that would mean that for at least half of the days in a month the "TNO" emissions are used. Given the regional differences between the "TNO" and "NH3SAT" inventories, wouldn't it be logical to assume some kind of persistency here? Thus, use the latest "NH3SAT" value if for some day there is no super-obs available? Also temporal interpolation would be an option. How would that change the emission totals? And the conclusions on temporal variability, e.g. at line 433?

We indeed could assume some persistency. As a test, we calculate a mean daily corrective factor for each region A, B, C, D and we applied this corrective factor at the grid-cell scale when IASI super-observations are not available. This results in the daily variability shown in light orange below, showing for example higher emissions in the northeastern part of France the second week of March. Nevertheless, the small number of IASI pixels taken into account for this corrective factor may be partially contaminated and may not be sufficiently robust. We prefer not to integer this test in our study. In future work, variational inversion might be a better option to better take into account transport, chemistry on the one hand and adjust the window to reduce the impact of days with few data on the other hand.

---

## Author Comment (AC2) · 7 Sep 2020

**Reviewer #2**
**We wish to thank the referee for his/her helpful comments. The full reviews are copied hereafter and our responses are inserted. The comments of the reviewer are in normal black and our answers in bold.**

The authors present an inter-comparison of two alternative inventories with the European reference inventory (TNO) to quantify the French NH3 emissions during spring 2011; (i) NH3SAT inventory which is based on a top-down approach of correcting TNO-NH3 emissions based on total column observations and (ii) CADASTRE-CIT inventory which is built from the bottom-up based on modeled NH3 emissions related to fertilizer application and animal husbandry. There is a thorough comparison of inventories based on different regions of France with known anthropogenic NH3 emitting activities. The work specifies which regions in France, represented in the European reference inventory, need to be improved upon and highlights the general conclusion of improving NH3 emissions inventories based on measurements and process knowledge is required. Inventories are key inputs in forecasting air quality, so the goal of this study to build a more representative inventory over France is important work. Building NH3 inventories is very challenging because of its varied lifetime in the atmosphere and complex exchange mechanisms making its spatial distribution and temporal behavior difficult to predict. Atmospheric NH3 is an important precursor to PM and can also damage N-sensitive ecosystems, therefore, refined emissions inventories are needed for air quality modeling and monitoring emissions reductions. This study details methods to refine inventories, therefore, I would recommend publishing this manuscript after some revisions.

Major Comments
There have been other global inventories built by the atmospheric community using similar methods that have not been mentioned and would add to the discussion of the authors' work. Work from Zhang et al. (ACP, 18, 339-355, 2018), who reconcile bottom-up and top-down inventories, also show including more detailed information on crop-specific fertilizer application practices and met factors does a better job at reproducing spatial and seasonal variations in China, which seems to be similar in this study, but in France.
**We agree, we now mention the work of Zhang et al. [2018] in the introduction**.

How do the two alternative inventories compare to currently available inventories, aside from TNO? How does CADASTRE-CIT compare with the MASAGE_NH3 inventory? and other global inventories that represent France, such as EDGAR?

**First, a reference to the interesting study of Paulot et al., [2014] is now done in the introduction.**

**We have found a comparison of NH₃ European annual emissions between various inventories in Riddick et al., [2016]. Nevertheless, this comparison for annual budgets could be not relevant as we only focused on spring period. Nevertheless, we have added a sentence about the MASAGE_NH₃ inventory in Section 3.2: "The northeastern part of France presents the largest difference with the TNO-GEN inventory (48 ktNH₃) for both NH₃SAT and CADASTRE-CIT inventories (65 and 135 ktNH₃, respectively). The high emissions in the northeastern part of France are in agreement with the MASAGE_NH₃ inventory [Paulot et al., 2014], the magnitude of annual NH₃ emissions from mineral fertilizer being calculated by combining an inventory of crop**

**acreages, crop- and country-specific fertilizer application rates and fertilizer-, crop-, and application-specific emission factors."**

Riddick, S., Ward, D., Hess, P., Mahowald, N., Massad, R., and Holland, E.: Estimate of changes in agricultural terrestrial nitrogen pathways and ammonia emissions from 1850 to present in the Community Earth System Model, Biogeosciences, 13, 3397–3426, https://doi.org/10.5194/bg-13-3397-2016, 2016.

Is TNO built upon any of these inventories already?
**TNO is built upon official reported emissions by country under the Convention for Long-Range Transboundary Air Pollution (CLRTAP) from the Centre for Emission Inventories and Projections (CEIP, http://www.ceip.at/webdab-emission-database/oficially-reported-emission-data/). If these data were not available or not of sufficient quality, emissions data were replaced by emissions from the IIASA GAINS model (http://gains.iiasa.ac.at/models/gains_models.html). This especially applies to countries outside of the EU but that are part of the United Nations Economic Commission for Europe (UNECE) domain. In addition, JRC EDGAR data (http://edgar.jrc.ec.europa.eu/) have been used for gapfilling for countries that are part of the domain but not part of UNECE (i.e., Armenia, Azerbaijan, Georgia).**

**The full overview of the choices made per country by the TNO team can be seen in the supplementary materials of Kuenen et al. [2014].**

**Kuenen, J. J. P., Visschedijk, A. J. H., Jozwicka, M., and Denier van der Gon, H. A. C.: TNO-GEN-MACC_II emission inventory; a multi-year (2003–2009) consistent high-resolution European emission inventory for air quality modelling, Atmos. Chem. Phys., 14, 10963-10976, https://doi.org/10.5194/acp-14-10963-2014, 2014.**

Atmospheric NH3 is known to undergo bi-directional exchange with surfaces and this aspect is not discussed. The NH3SAT is generated based on chemical transport modeling of TNO, however, the CHIMERE model parameterizes NH3 dry deposition uni-directionally. Is this a limit of the model? If NH3 dry deposition is assumed to be a net sink, then in most cases that would underestimate atmospheric concentrations. How does this impact NH3 estimates? and to what degree?
**Indeed, the misrepresentation of deposition could have impact on our simulated NH$_3$ columns. We have added the following discussion in the CHIMERE description: « As most of the models in the world, the parameterization of NH$_3$ dry deposition is unidirectional in CHIMERE. The parameterization of a bidirectional exchange with surfaces in Wichink Kruit et al. [2012] increased their yearly mean modeled LOTOS-EUROS European ammonia concentrations almost everywhere, and particularly over agricultural source areas. However, Zhu et al. [2015], with the Goddard Earth Observing System-Chemistry (GEOS-Chem) global CTM, estimated decrease of NH$_3$ European concentrations in April, when the inclusion of a compensation point for vegetation is included. Further work needs to be done to better investigate the sensitivity of NH$_3$ concentrations to the bi-directional exchange for dry deposition. Nevertheless, without such parameterization for bi-directional exchange, Azouz et al. [2019] assessed that regional models such as CHIMERE usually operating with large grid cell sizes simulate quite well the average NH$_3$ dry deposition flux over a large domain of simulation. »**

Wichink Kruit, R. J., M. Schaap, F. J. Sauter, M. C. van Zanten, and W. A. J. van Pul: Modeling the distribution of ammonia across Europe including bi-directional surface-atmosphere exchange, Biogeosciences, 9, 5261–5277, doi:10.5194/bg-9-5261-2012, 2012.

Zhu, L., D. Henze, J. Bash, G.-R. Jeong, K. Cady-Pereira, M. Shephard, M. Luo, F. Paulot, and S. Capps: Global evaluation of ammonia bidirectional exchange and livestock diurnal variation schemes, Atmos. Chem. Phys., 15, 12,823–12,843, doi:10.5194/acp-15-12823-2015, 2015.

Azouz, N. et al: Comparison of spatial patterns of ammonia concentration and dry deposition flux between a regional Eulerian chemistry-transport model and a local Gaussian plume model, Air Quality, Atmosphere & Health (2019) 12:719–729 https://doi.org/10.1007/s11869-019-00691-y, 2019.

Can the authors comment on how well CHIMERE can predict particulate NH4+? How much does that influence NH3 estimates?

**We wanted to evaluate the different sets of emissions by comparison with independent ammonium NH$_4^+$ surface measurements. Among the nine available stations presenting NH$_4^+$ measurements during the spring 2011, only one site is exploitable, as it presents a significant number of measurements for each month in the spring 2011. This station is located at Rouen (FR25048), in the northeastern part of France (e.g., in the region Haute-Normandie). To our knowledge, there is no additional interpretable NH$_4^+$ surface measurements for the focused period here, making the interpretation of the results difficult and this is the reason why we did not add this evaluation in the study. In Rouen, the ammonium measurements presented a strong maximum in March, and a decrease in April and May. The daily variability was well reproduced by simulations with all three inventories, even if the simulations often underestimate the NH$_4^+$ maximums.**

**NH$_4^+$ comparisons during other periods are scarce also. For instance, Tuccella et al., [2019] compared CHIMERE simulated and observed NH$_4^+$ at the Cabaux supersite and found average concentrations for May 2008 of 1.3 µg/m$^3$ for both, with a correlation coefficient of 0.52. For the Paris agglomeration between September 2009 and 2010, the modelled regional NH$_4^+$ burden was 1.8 µg/m$^3$ while the modelled one was 1.6 µg/m$^3$ [Petetin et al., 2016]. From June to September 2010, 83% of modelled total NH$_x$ was gaseous, while in the model, it was only 50%, coherent with this NH$_3$ was underestimated especially during warmer days. Thus, it is concluded for one site and season, that particulate NH$_4^+$ has a low to medium impact on NH$_3$.**

**We have added the following text in the CHIMERE description in Section 2.2.1: "The evaluation of CHIMERE NH$_3$ and NH4+ concentrations should be done against NH$_3$ (as done in Fortems-Cheiney et al., [2016]) and NH4+ measurements. Nevertheless, to our knowledge, there is no available NH$_3$ measurement over France for the focused period here. There is interpretable NH$_4^+$ surface measurements at only one site, making the interpretation of the results difficult. NH$_3$ and NH$_4^+$ comparisons during other periods are scarce also. For instance, Tuccella et al., [2019] compared CHIMERE simulated and observed NH$_4^+$ at the Cabaux supersite and found average concentrations for May 2008 of 1.3 µg/m$^3$ for both, with a correlation coefficient of 0.52. For the Paris agglomeration between September 2009 and 2010, the modelled regional NH$_4^+$ burden was 1.8 µg/m$^3$ while the modelled one was 1.6 µg/m$^3$ [Petetin et al., 2016]. From June to September 2010,**

**83% of modelled total NH$_x$ was gaseous, while in the model, it was only 50%, coherent with this NH$_3$ was underestimated especially during warmer days. Thus, it is concluded for one site and season, that particulate NH$_4^+$ has a low to medium impact on NH$_3$."**

Petetin, H., Sciare, J., Bressi, M., Gros, V., Rosso, A., Sanchez, O., Sarda-Estève, R., Petit, J.-E., and Beekmann, M.: Assessing the ammonium nitrate formation regime in the Paris megacity and its representation in the CHIMERE model, Atmos. Chem. Phys., 16, 10419–10440, https://doi.org/10.5194/acp-16-10419-2016, 2016.

Tuccella, P.; Menut, L.; Briant, R.; Deroubaix, A.; Khvorostyanov, D.; Mailler, S.; Siour, G.; Turquety, S. Implementation of Aerosol-Cloud Interaction within WRF-CHIMERE Online Coupled Model: Evaluation and Investigation of the Indirect Radiative Effect from Anthropogenic Emission Reduction on the Benelux Union. *Atmosphere* 2019, *10*, 20, 2019.

What are the limits of the Volt'air model? It is usually used to predict emissions from slurry applications, so it doesn't account for a crop canopy. Does that matter? Are there any fast-growing crops that would sprout in the first month in which the model was run?

**Volt'Air indeed does not account for the canopy effect on NH$_3$ volatilization. Well-developed canopies reduce soil surface temperature and wind speed in the canopy i.e., the rate of vertical NH$_3$ transport from the soil surface. Growing canopies also absorb the NH$_3$ gas emitted by the mineral fertilizer or manure, in large quantities for well-developed canopies. In the case of applications to the soil surface beneath the crop canopy, the use of Volt'Air would lead to an overestimation of emissions for fertilizations occurring during plant growth, depending on the type, the height or the leaf area index, and the phenological stage of the crop. This would be mainly the case for slurry applied using either trailing hose or trailing shoe. But, in France in 2010-11, (i) manure applications on arable crops occurred mainly before sowing, i.e., on bare soils; (ii) band spreading techniques were not of wide use, and anyway; (iii) in practice, grassland fertilization is most often carried out immediately after the grass is cut when it is in need of mineral N i.e., when the canopy has no effect on ammonia volatilization. Furthermore, when fertilizers and manure are applied on well-developed canopies, part of them may coat the crop or grass leaves, partially enhancing exchange surface with air. Volatilization is not that much reduced in this case. That is why in a first approach, we used Volt'Air for bare soils.**

What is the availability of ground-based NH3 measurements in regions which have the most variation? If so, how do they compare with IASI observations if we assume the total column of NH3 is all at the surface? Were ground-based NH3 concentrations used as an a priori for IASI total column calculations?

**To our knowledge, there is no ground-based NH$_3$ measurements available that could have allowed an independent evaluation of our results in regions with highest and most variable NH$_3$ concentrations Nevertheless, we have added a reference to the study of Tournadre et al, [2020] in Section 3.3: "This maximum in March is also noticed by Tournadre et al. [2020], providing nine years of total column observations from ground-based infrared remote sensing over the Paris megacity."**

Tournadre, B., Chelin, P., Ray, M., Cuesta, J., Kutzner, R. D., Landsheere, X., Fortems-Cheiney, A., Flaud, J.-M., Hase, F., Blumenstock, T., Orphal, J., Viatte, C., and Camy-

**Peyret, C.: Atmospheric ammonia (NH₃) over the Paris megacity: 9 years of total column observations from ground-based infrared remote sensing, Atmos. Meas. Tech., 13, 3923–3937, https://doi.org/10.5194/amt-13-3923-2020, 2020.**

**Ground-based NH₃ measurements are not used as an a priori for IASI total column calculations.**

Minor Edits
Line 401 is missing a parenthesis - (68, 73, and 71 ktNH3, respectively).
**It has been corrected.**

I am not sure if there is something funky with the text editor file that messes with the spacing in the pdf, but there are some words that have been compounded throughout the article.
**We apologized for the inconvenience. It has been corrected.**

IASI reference should also include Clarisse, L., Clerbaux, C., Dentener, F., Hurtmans,D., and Coheur, P.: Global ammonia distribution derived from infrared satellite observations, Nat. Geosci., 2, 479–483, 2009.
**The reference has been added.**

---

## Author Comment (AC3) · 7 Sep 2020

**Reviewer #3**
**We wish to thank the referee for his/her helpful comments. The full reviews are copied hereafter and our responses are inserted. The comments of the reviewer are in normal black and our answers in bold.**

GENERAL COMMENTS
The manuscript submitted by Fortems-Cheiney et al. assesses different methods to determine NH3 emissions that are used in the scientific community. The authors compare three state-of-the art approaches for NH3 emission retrievals and discuss their spatial and temporal differences over a period in Spring 2011, which also includes the time of fertilizer application. While the analysis was performed for France, the discussed discrepancies between methods is also of high interest to researcher focusing on other regions. Due to the general high uncertainties in NH3 emission estimates, the presented study is of high relevance for improving chemical transport models. The manuscripts is well and clearly written and the analysis and interpretation of results is sound. Given the relevance of the topic for air quality, nitrogen deposition and atmospheric chemistry in general, I suggest approving the manuscript for publication in Atmospheric Chemistry and Physics after addressing the following comments.

SPECIFIC COMMENTS
L. 34-35: I suggest to clarify that the given values are the cumulative emissions from March to May. It is mentioned above, however, it would be good to restate here to not confuse it with an emissions flux.
**We have changed the sentences to clarify this: "The total spring budgets, from March to May 2011, at the national level are higher when calculated with both alternative inventories than with the reference one, the difference being more marked with CADASTRE-CIT. NH$_3$SAT and CADASTRE-CIT inventories both yield to large NH$_3$ spring emissions due to fertilization on soils with high pH in the northeastern part of France".**

L. 123: I assume the study period was from 1 March to 31 May? Still, the day should be included in the period description.
**The days have been included.**

L. 217-219: Above it is stated that only morning overpass measurements are used. This may introduce a systematic bias to the NH3SAT emissions as emissions may be larger during midday due to higher temperatures or increased activity (e.g. mobile emissions). If I understand it correctly, the effect also depends on the TNO-GEN input data for CHIMERE, i.e. if monthly or a diurnal profile is applied to that. While I agree with the decision to only use morning overpass data, this potential bias should be explained and how it may impact the findings (e.g. could it explain why NH3SAT emissions are lower than those of CADASTRE-CIT in northeast, where fertilizer emission are high?).
**Indeed, only morning overpass measurements are used, when the emissions are not at their maximum. These morning IASI data are compared with NH$_3$ simulated columns at the same hours in the morning, using the diurnal profile used in the TNO-GEN inventory. This is introducing an uncertainty to the NH$_3$SAT emissions, but not a systematic bias.**

**One perspective for this question would be an inter-comparison using both IASI and CrIS (with an overpass in the beginning of the afternoon). Unfortunately, CrIS was not flying during spring 2011, our focused period.**

L. 219-221: Could one also take only those periods where the IASI quality criteria are met and omit the other periods for the analysis instead of using the TNO-GEN emissions? How would that change the average correction factors presented in this study?
**As a test, we calculated a mean daily corrective factor for each region A, B, C, D and we applied this corrective factor at the grid-cell scale when IASI super-observations are not available. This results in the daily variability shown in light orange below, showing for**

**example higher emissions in the northeastern part of France the second week of March. Nevertheless, the small number of IASI pixels taken into account for this corrective factor may be partially contaminated and may not be sufficiently robust. We prefer not to integer this test in our study.**

L. 246-247: From this paragraph alone it is not clear whether the manure field spreading is included the organic fertilizer application, which I would assume. If that is the case, you may add here "..., which is part of the organic fertilizer emissions" or similar.

**The formulation is possibly incorrect: all husbandry effluents are considered, whether they are liquid or solid, what we thought was contained in the term "manure" as opposed to "farm yard manure". The authors have modified the sentence to remove any ambiguity into:**

**"For livestock emissions, with the exception of the stage of effluent spreading in the field, the less detailed inventory of the French Interprofessional Technical Centre for Studies on Air Pollution CITEPA is used."**

L. 314: In the Results & Discussions section the authors describe the differences between the different approaches and also give here and there some suggestions why these differences occur. Still the discussions can be extended at the end of the section, e.g. by elaborating more on the uncertainties of each approach, leading to recommendations for future research directions and at which part each of the approaches should be improved (e.g. need for implementation of bidirectional exchange module in CHIMERE?). Although the paper is targeted on France, the findings are also insightful for users of NH3 emission estimates in other parts of the world. Therefore, relating the findings in the discussion also to inventories/approaches used in other regions would in my opinion make the manuscript stronger and attract more interest by a wider community.

**Thank you for this suggestion, which allows highlighting the implications of our study. As already answered to referee 1, we added a couple of sentences at the end of the Conclusion section:**

**"Yet, current results of our study have important implications for air quality modelling over Europe. The important changes in the spatial distribution of $NH_3$ emissions as a function of soil properties is of general concern not only for France, but for whole Europe. Soils are alkaline or neutral (pH>6) not only over North-Eastern France, but also over large parts of Italy, eastern Spain, or eastern Germany [Reuter, 2008]. Over these regions, our study suggests potentially larger $NH_3$ emissions than with a constant emission factor treatment, with impacts then on fine particle formation. These features should be included in "operational" emission inventories used for air quality modelling."**

**Reuter, H.I., Lado, L.R., Hengl, T. and Montanarella, L.: Continental-scale digital soil mapping using European soil profile data: soil pH, Hamburger Beiträge zur Physischen Geographie und Landschaftsökologie – 92 Heft 19/2008, pp. 91-102, 2008.**

L. 428: I suggest restating here in short the hypothesis from the introduction.
**We rather have removed the reference to the hypothesis from the introduction in this sentence.**

L. 405-408: Would market gardening be included in the TNO-GEN inventory? If not, an underestimation in that region would also apply to the TNO-GEN inventory.

**To our knowledge, market gardening is indeed not included in the TNO-GEN inventory. We have changed the sentences: "Over the southeastern part of France, CADASTRE-CIT is about 23% lower than NH₃SAT (28 and 37 ktNH₃, respectively, Table 2). One hypothesis to explain the lower $NH_3$ emissions in CADASTRE-CIT is that market gardening is**

**important in this area and not taken into account in the CADASTRE-CIT inventory [Ramanantenasoa et al., 2018; Génermont et al., 2018]. Nevertheless, market gardening is not included, to our knowledge, in the TNO-GEN inventory. TNO-GEN and NH₃SAT inventories being in quite good agreement in terms of budget (35 and 37 ktNH₃, respectively, Table 2), further work is required to understand these discrepancies**."

TECHNICAL COMMENTS
Some words are accidentally merged, for example in lines 38, 50, 51, 93, 139, 196,197, 207, 209, 210, 211, 272, 287, 291, 320, 371, 381, 400, 433, 466, 475 and 478.
**We apologized for the inconvenience. It has been corrected.**

The description and spelling of regions like "North-Eastern France" vs. "northeastern France" should be consistent throughout the manuscript.
**This has been corrected and the description of the different regions are now consistent throughout the manuscript.**

L. 26: The sentence structure can be improved as it can be misleading what the actual methods are which the authors refer to. For example, I suggest inserting ":", "namely"...or something similar after "emissions".
**The sentence has been changed: "In this study, we compare NH₃ emissions in France during the spring 2011 from one reference inventory, the TNO inventory, and two alternative inventories that account in different manners for both the spatial and temporal variabilities of the emissions: (i) the NH₃SAT satellite-derived inventory based on IASI NH₃ columns and (ii) the CADASTRE-CIT inventory that combines NH₃ emissions due to nitrogen fertilization calculated with the mechanistic model VOLT'AIR on the database of the CADASTRE_NH₃ framework and other source emissions from the CITEPA."**

L. 292: I am not familiar with this terminology but there might be better terms than "desegregation" and "reagregation" in English.
**"Desegregation" and "reagregation" are dedicated terms that refers to the fact of distributing the emissions obtained at an initial scale to another entity and then reallocating them to different scales (here to grids compatible with the CHIMERE model, according to the distribution of crops in that grid cell). We do not know alternative terms and we kept them in the text.**

L. 379: I suggest using "In bold are marked...".
**We have changed the sentence: « Regions for which the inventories NH₃SAT and CADASTRE-CIT propose the same sign of relative differences are marked in bold"**

L. 401: A closing parenthesis is missing.
**It has been corrected.**